# Enhancement of Mycelial Growth and Antifungal Activity by Combining Fermentation Optimization and Genetic Engineering in *Streptomyces pratensis* S10

**DOI:** 10.3390/microorganisms13081943

**Published:** 2025-08-20

**Authors:** Lifang Hu, Yan Sun, Ruimin Jia, Xiaomin Dong, Xihui Shen, Yang Wang

**Affiliations:** 1College of Plant Protection, Northwest A&F University, Yangling, Xianyang 712100, China; hufang@nwafu.edu.cn (L.H.); sunyan@nwafu.edu.cn (Y.S.); jiaruimin@nwafu.edu.cn (R.J.); dongxiaomin@nwafu.edu.cn (X.D.); 2State Key Laboratory for Crop Stress Resistance and High Efficiency Production, Northwest A&F University, Yangling, Xianyang 712100, China; 3Shaanxi Key Laboratory of Agricultural and Environmental Microbiology, College of Life Science, Northwest A&F University, Yangling, Xianyang 712100, China

**Keywords:** *Streptomyces pratensis* S10, antifungal activity, fermentation optimization, Plackett–Burman design, response surface methodology, gene deletion

## Abstract

The biocontrol strain *Streptomyces pratensis* S10 was isolated from tomato leaf mold. The fermentation broth of strain S10 can effectively control Fusarium head blight (FHB), caused by *Fusarium graminearum*. Enhancing antifungal activity is essential in advancing its commercialization. In this study, we aimed to improve the antifungal activity of S10 by integrating fermentation optimization and genetic engineering. Single-factor experiments revealed that seven parameters, namely corn flour, yeast extract, NaNO_3_, CaCO_3_, K_2_HPO_4_, KCl, ZnSO_4_·7H_2_O, and MnCl_2_·4H_2_O, were identified as significant components. A Plackett–Burman design (PDB) indicated that corn flour, yeast extract, and ZnSO_4_·7H_2_O were the most critical variables affecting its inhibitory activity and mycelial biomass. The fermentation medium was further determined based on the steepest climbing experiment and a Box–Behnken design (BBD), and the mycelial dry weight of *S. pratensis* S10 was improved from 2.13 g/L in Gauze’s synthetic No. 1 medium to 8.12 g/L in the optimized medium, closely aligning with the predicted value of 7.98 g/L. Under the optimized medium, the antifungal rate of *F. graminearum* increased from 67.36 to 82.2%. The spore suspension of strain S10 cultured in the optimized medium substantially improved its biocontrol efficacy against FHB. Moreover, disruption of the key gene *tetR* led to increased antifungal activity of strain S10 against *F. graminearum*. Importantly, the antifungal activity of Δ*tetR* was greatly increased under the optimized fermentation medium. This study suggests that the gene *tetR* negatively regulates bioactive compound biosynthesis, and the optimized medium provides favorable conditions for the growth of S10. These observations establish an extended basis for the large-scale bioactive metabolite secretion of *S. pratensis* S10, providing a strong foundation for sustainable FHB management in agriculture.

## 1. Introduction

Food production must increase by approximately 60% to feed the estimated global population of 10 billion by 2050 [1]. Achieving this goal necessitates not only enhanced crop yields but also substantial reductions in yield losses attributable to pathogens and pests. Recently, an expert assessment revealed that pathogens and pests collectively cause average yield losses of 22.5% for five major crops, namely wheat, rice, maize, potato, and soybean [2]. Wheat, being the primary crop of approximately 35% of the global population, is particularly vulnerable to fungal disease, including blotches, rusts, and Fusarium head blight (FHB), which collectively account for 15–20% of the annual yield losses [3,4]. For instance, the annual average incidence of FHB, predominantly triggered by *Fusarium graminearum* in China, impacts over 4.5 million hectares, accounting for roughly 20% of the total wheat cultivation area, and has resulted in severe yield production declines [5]. Moreover, *F. graminearum* secretes various mycotoxins, like deoxynivalenol (DON) and zearalenone, contaminating grains and being hazardous to humans and animals [6]. Currently, chemical fungicides are the main approach to controlling FHB. However, the long-term intensive application of fungicides has led to the emergence of resistant *F. graminearum* isolates in fields [7]. Importantly, the application of several fungicides at certain levels stimulates mycotoxin biosynthesis [8,9]. Consequently, it is imperative to develop novel, sustainable strategies to control FHB and mitigate mycotoxin accumulation.

Several strategies have been applied to manage FHB, including host resistance, cultural practices, biological control, and fungicide application. Among these, biological control based on microorganisms has attracted significant agricultural interest due to its capacity to selectively target fungal infections while reducing negative environmental issues [10]. The mechanisms of biological control include competition, antifungal substance production, lysis and parasitism, the induction of plant growth, and plant resistance [11]. Extensive studies have demonstrated that an effective reduction in FHB severity can be achieved by bacterial strains, fungal biocontrol agents (BCAs), and *Streptomyces* species [12,13,14]. Within BCAs, *Streptomyces* species are remarkable for their extraordinary capacity to secrete a vast array of metabolites, such as niphimycin C and frenolicin B, which have been studied for their potential as biofungicides [15,16,17]. As the most abundant genes of actinomycetes, *Streptomyces* spp. are an essential source of secondary metabolites. More than 75% of bioactive substances are derived from *Streptomyces* species [18]. Thus, *Streptomyces* spp. have received considerable attention in agriculture and have become an important source of biopesticides. However, the low production rates of bioactive substances in *Streptomyces* spp. have led to high costs and seriously restricted the application of these substances in agriculture. Hence, enhancing the production of bioactive compounds is essential to expand their practical application.

Fermentation optimization and strain improvement are key strategies applied to boost the yields of bioactive compounds. To achieve higher metabolite production, the optimal fermentation medium and conditions needed to be determined. Statistical methods are widely employed to evaluate parameter interactions and determine optimal conditions [8,19]. Initially, the Plackett–Burman design (PBD) enables the efficient screening of significant factors at two levels [20]. Subsequently, the crucial parameters identified via the PBD can undergo refinement using the Box–Behnken design (BBD) and response surface methodology (RSM) [8,16]. The RSM is a statistical–mathematical experimental technique applied to identify the optimal parameters and conditions of fermentation processes [8]. The RSM utilizes statistically driven experimental designs to construct quadratic polynomial models that elucidate variable relationships [21]. By comprehensively accounting for variable interactions, the RSM is extensively applied to optimize medium compositions and culture conditions, thereby enhancing the production efficiency of bioactive compounds [22,23,24]. Furthermore, the optimized fermentation medium, validated through the RSM, provides a robust basis for industrial-scale production. Notably, the integration of these optimization strategies has been successfully implemented in diverse microbial systems, including *Streptomyces* spp., bacteria, and fungi, leading to significant improvements in bioactive metabolite synthesis [19,25,26]. For instance, the dry weight of mycelia in *S. alfalfa* XN-04 was increased 7.74-fold over the original medium via the above methods [16]. Additionally, genetic engineering strategies and natural strain screening are efficient approaches to enhancing antifungal activity [27]. LrhA was identified as a negative transcriptional regulator in *Pantoea agglomerans* ZJU23. The antifungal activity of Δ*LrhA* was significantly enhanced in comparison with the wild-type strain [8].

In our previous study, we demonstrated that *Streptomyces pratensis* S10, a biocontrol agent isolated from tomato leaf mold, exhibited remarkable biocontrol efficacy against FHB and effectively reduced DON production [11]. However, the large-scale application of strain S10 has been hindered by its low biomass yield and prolonged fermentation period. To address these limitations, this study aimed to screen the significant variables influencing the growth of strain S10 from various parameters through the PBD assay. The central points of the BBD were examined via the steepest climbing assay. Ultimately, the optimal medium composition was established with the RSM. Moreover, a key gene, the *tetR* mutant, was constructed, which negatively regulates the growth and antifungal activity of strain S10. This study provides a theoretical basis for the development and application of *S. pratensis* S10 as an effective microbial fertilizer.

## 2. Materials and Methods

### 2.1. Strains, Plasmids, and Culture Conditions

Spores of the biocontrol strain *S. pratensis* S10 (stored by the Plant Disease Biological Control Laboratory of Northwest A&F University, Xianyang, China) were produced by culturing on solid mannitol soybean agar (MS) for 7 days at 28 °C in the dark. *Fusarium graminearum* strain PH-1, a phytopathogenic fungus, was deposited at the Agricultural Research Service Culture Collection (NRRL 31804) and was grown at 25 °C on potato dextrose agar (PDA). *Escherichia coli* DH5α was employed for DNA cloning, and intergeneric conjugation using *E. coli* ET12567/pUZ8002 facilitated plasmid transfer into strain S10. pKC1139 was employed for gene knockout in *Streptomyces* species. The mutant was complemented with pSETC harboring the strong constitutive expression promoter P_SF14_.

### 2.2. Preparation of Seed Fermentation Broth

Six spore cakes (6 mm) of *S. pratensis* S10 were transferred to 100 mL of Gauze’s synthetic No. 1 (GS) medium in 250 mL Erlenmeyer flasks for inoculation preparation. Then, the cultures were cultivated at 28 °C with shaking at 180 rpm for 72 h and then utilized as seed inoculation for subsequent experiments.

### 2.3. Single-Factor Experiment

Single-factor experiments were employed to select the optimal medium composition. The fermentation medium composition was based on GS medium (20 g soluble starch, 0.5 g NaCl, 1 g KNO_3_, 0.5 g K_2_HPO_4_, 0.5 g MgSO_4_·7H_2_O, 0.01 g FeSO_4_·7H_2_O, and 15 g agar per liter). To screen carbon sources, glucose, maltose, sucrose, corn flour, fructose, galactose, oats, and soluble starch were individually tested at the same concentrations, replacing the original carbon source while keeping other components constant. For nitrogen source selection, 1 g/L of KNO_3_ was substituted with equivalent concentrations of (NH_4_)_2_SO_4_, NH_4_NO_3_, NaNO_3_, yeast extract, beef extract, soybean cake powder, and peptone, with all other ingredients unchanged. Mineral salts were optimized by evaluating K_2_HPO_4_, MgSO_4_·7H_2_O, CaCO_3_, KCl, KH_2_PO_4_, and NaCl at 0.5 g/L. Additionally, trace elements including CuSO_4_·5H_2_O, CoCl_2_·6H_2_O, MnCl_2_·4H_2_O, FeSO_4_·7H_2_O, and ZnSO_4_·7H_2_O at 0.01 g/L were assessed. The experiments were performed in triplicate. Following fermentation, the antifungal activity of the mycelium extract against *F. graminearum* was examined, as described in the Appendix A. Additionally, the fermentation culture was collected to determine the dry weight of the mycelial biomass as a measure of fungal growth. Generally, in non-growth-associated fermentation, secondary metabolite production is substantially proportional to the quantity of biomass [16]. Therefore, it is essential to increase the total biomass of microorganisms, especially *S. pratensis* S10, since its antifungal metabolites are produced inside the cells. Taking into account the above, we focused on the optimization of medium components, seeking a substantial increase in biomass without reducing the production of secondary metabolites. Therefore, the mycelial dry weight was utilized as the response in subsequent experiment designs. The details of the single-factor concentration screening test are presented in the Appendix A.

### 2.4. Response Surface Optimization Experiment

The response surface methodology (RSM) is a well-established statistical tool employed to optimize intricate processes via modeling and examine the interactions among multiple independent factors and a single response. The optimization of medium components for *S. pratensis* S10 in this study was performed using the RSM, with mycelial biomass serving as the response variable. The optimization proceeded through three consecutive steps. Initially, a Plackett–Burman design (PBD) assay was performed to screen out the factors with significant effects. Then, a steepest climbing test was carried out to move toward the optimal regions of these factors. Finally, a Box–Behnken design was utilized for the modeling and refinement of the optimal medium components. Each step is elaborated in the subsequent subsections.

#### 2.4.1. Screening of Medium Components of S10 by Plackett–Burman Design Assay

Based on preliminary single-factor tests, a Plackett–Burman design (PBD) was employed to identify crucial medium ingredients influencing the mycelium biomass and antifungal activity of strain S10. Twelve runs with seven parameters (A-G) at two levels (−1 and +1) were examined according to the single-factor assays, as detailed in Appendix A. Seven different variables were taken into account throughout the experiments, namely soluble starch, NaNO_3_, soybean cake powder, K_2_HPO_4_, ZnSO_4_·7H_2_O, CaCO_3_, and MnCl_2_·4H_2_O. Four dummy variables were incorporated into the 12 experiments to estimate the standard error. The data were processed and analyzed via the Design-Expert software (Version 10.0, Stat-Ease, Inc., Minneapolis, MN, USA). All experiments were conducted in triplicate, and the mean mycelial dry weight was the response variable. The effects of medium constituents on the mycelial dry weight were evaluated via analyzing *p* values derived from an analysis of variance (ANOVA). Factors were deemed statistically significant if their *p* value (Prob > *F*) was less than 0.05.

#### 2.4.2. Steepest Climbing Test

The steepest climbing test was performed according to the key influencing parameters identified from the PBD experiment results. The step length and direction were identified by considering the effect coefficient of each parameter and practical constraints. The resulting optimal region from this process defined the central points for the Box–Behnken design.

#### 2.4.3. Box–Behnken Design (BBD) Experiment

Applying the parameters and concentrations identified from the PBD assay and the steepest climbing experiment, a Box–Behnken design (BBD) was employed to evaluate the effects of corn flour (A), yeast extract (B), and ZnSO_4_·7H_2_O (C) on antifungal activity and mycelium biomass. Each factor was tested at three levels: −1 (low), 0 (middle), and +1 (high), as detailed in Appendix A. The reliability of this model was assessed using Student’s *t* test. Parameters for liner and quadratic empirical models were estimated from BBD data via multiple linear regression analysis using the least squares method. Model coefficients, R^2^ values, *F* values, and significance probabilities were predicted via the Design-Expert software, Version 10.0 (Stat-Ease, Inc., Minneapolis, MN, USA), validating the statistical significance of the independent variable [28]. A variance analysis of the regression model was performed to assess the impact of each individual parameter and their interactions on the response variable. The optimal medium composition was determined through resolving the ternary quadratic equation and evaluating response surface contour plots. Following the prediction of the optimal parameters, validation experiments were carried out via fermentation in an optimized medium.

To evaluate the accuracy of this model and validate the predicted results, experimental verification was performed by fermentation in a medium with an optimized composition, followed by a comparative analysis between the experimental and predicted response values.

### 2.5. Evaluation of Biocontrol Efficacy

Experiments were conducted to evaluate the biocontrol efficacy of the fermentation suspension generated by S10 in GS and an optimized medium against Fusarium head blight. Wheat plants at the 10% anthesis stage, with 15 cm stems, were grown under natural conditions and collected from the Caoxinzhuang Agroecosystem Experimental Station, Yangling, China (108°05′28′′ E, 34°18′06′′ N) and transplanted into 20 cm pots (5 plants/pot) filled with a sterilized 2:1 volume ratio of peat to vermiculite. Plants were maintained in a greenhouse at 25 °C under a 16 h light/8 h dark photoperiod. At flowering, wheat spikes were sprayed with a spore suspension (1 × 10^8^ CFU mL^−1^) of S10 using an atomizer. Subsequently, conidial suspensions of *F. graminearum* (10 μL, 1 × 10^5^ mL^−1^) were inoculated into a central floret per spikelet. Sterile distilled water (10 μL) served as the control inoculation. Following inoculation, the wheat heads were covered with plastic bags for 48 h to sustain high humidity. After 14 days of inoculation, the disease index and the diseased spikelets were assessed following the method previously described [7]. The experiment was carried out three times independently, with each replicate consisting of at least of 10 wheat heads.

### 2.6. Construction of tetR Deletion Mutant and Complementation

A negative regulator gene, *tetR* (Gene ID: gene3686), was in the predicted bioactive substance biosynthesis gene cluster. To construct the *tetR* deletion mutant, upstream and downstream regions were amplified by PCR from the genomic DNA of strain S10 using the primers ArmF/ArmR and BrmF/BrmR (Appendix A). The gentamicin resistance gene was amplified from pBR1MCS-5 with the Gm-F/Gm-R primer pair. These three fragments were ligated into the *Hind III* and *EcoR I* restriction enzyme (New England Biolabs, Inc., Ipswich, MA, USA) sites in pKC1139 via Gibson DNA assembly protocols [29]. The resulting construct, designated pKC1139-tetR, was introduced into strain S10 through intergeneric conjugation by GS medium, and double-crossover recombinants were chosen by MS medium using the reported method [27]. The mutant, designated Δ*tetR*, was verified by PCR using the tetF/tetR primer pair (Appendix A). PCR was conducted with Phusion DNA polymerase (New England Biolabs, Inc., Ipswich, MA, USA), and the cycling conditions were as follows: initial denaturation at 98 °C for 30 s, followed by 35 cycles of 98 °C for 10 s, 62 °C for 30 s, and 72 °C for 1 min (depending on the length of the gene sequence), and a final extension at 72 °C for 10 min.

To complement the *tetR* gene in Δ*tetR*, the gene fragment was amplified via PCR using primers ComF and ComR (Appendix A). The resulting amplicon was subsequently ligated into the *BamH I* and *Spe I* restriction enzyme sites of plasmid pSETC. The constructed plasmid, designated pSETC-*tetR*, was next transferred to Δ*tetR* through intergeneric conjugation, as previously described. To confirm the complementation strain com-Δ*tetR*, PCR amplification of the apramycin resistance gene was performed.

### 2.7. Statistical Analysis

Three independent replicates were performed for all assays in this study. The results are shown as the mean ± standard deviation (s.d.). Differences between two groups were analyzed with Student’s *t* test. Multiple comparisons were evaluated with one-way analysis of variance (ANOVA) with Duncan’s test. Differences among treatment groups were statistically significant at *p* < 0.05. Experimental designs for the PBD and BBD were obtained via the Design-Expert software (Version 10.0, Stat-Ease, Inc., Minneapolis, MN, USA).

## 3. Results

### 3.1. Screening of Different Medium Components

The effects of diverse growth medium constituents, including nitrogen sources, carbon sources, mineral salts, and trace elements, on the biomass yield and antifungal activity of S10 were determined based on GS medium by a single-factor assay.

#### 3.1.1. Screening of Carbon Sources and Concentrations

The effects of eight various carbon sources on the biomass yield and antifungal activity at a final concentration of 20 g/L were investigated. The results showed that the medium supplemented with corn flour, galactose, and oats as the sole carbon source significantly enhanced the growth of strain S10 compared to other substrates (Figure 1A). The inhibition rate (%) was applied to reflect the yield of antifungal compounds. The inhibition rate was highest when corn flour used as the carbon source, followed by glucose and maltose (Figure 1E). Thus, corn flour was selected for further investigations. To identify the optimal concentration of corn flour, the basic medium was supplemented with it at concentrations of 0, 5, 10, 20, 40, and 80 g/L. The results indicated that the inhibition rate peaked at a 20 g/L corn flour concentration (Figure 2A). Nevertheless, the inhibitory activity declined as the concentration of corn flour increased (Figure 2A). Thus, 20 g/L corn flour was selected as the carbon source for subsequent investigations.

#### 3.1.2. Screening of Nitrogen Sources and Concentrations

Nitrogen sources play a vital role in microbial growth and the production of desired bioactive metabolites. We evaluated the effects of various alternative nitrogen sources on the biomass yield and antifungal activity by substituting KNO_3_ in the GS medium while keeping a constant concentration of nitrogen. The data demonstrated that soybean cake power and yeast extract significantly enhanced the mycelial growth of strain S10 in comparison with other nitrogen sources, with the mycelium biomass yields reaching 4.87 g/L and 4.73 g/L, respectively (Figure 1B). In addition, the media containing NaNO_3_ and NH_4_NO_3_ were more effective regarding the antifungal activity of S10, with the inhibition rates reaching 67.21% and 63.96%, respectively (Figure 1F). According to these results, NaNO_3_ and yeast extract were selected as the nitrogen sources for the further optimization of the fermentation medium. To determine the ideal concentrations of yeast extract and NaNO_3_, a series of assays were carried out with different concentrations added to the basic medium. The data showed that NaNO_3_ at 1 g/L and yeast extract at 5 g/L were necessary to achieve optimal antifungal activity, and the inhibition rate decreased as the concentrations exceeded these thresholds (Figure 2B,C).

#### 3.1.3. Screening of Mineral Salt Composition and Concentration

Mineral salts play a vital role as essential precursors in the biosynthesis of secondary metabolites. Consistently, our findings demonstrated that mineral salts significantly influenced antifungal compound production in *S. pratensis* S10. Among these, CaCO_3_ and KCl markedly increased the mycelial growth and antifungal activity of *S. pratensis* S10 (Figure 1C,G). The analysis of the optimal concentrations further revealed that the maximal effects were achieved at 0.5 g/L of CaCO_3_ and 2 g/L of KCl (Figure 2D,E).

#### 3.1.4. Trace Elements

The results demonstrated that the optimal biomass and rate of inhibition were attained at concentrations of 0.02 g/L MnCl_2_·4H_2_O and 0.02 g/L ZnSO_4_·7H_2_O (Figure 1D,H and Figure 2F,G). Based on these findings, these concentrations were selected for further experiments.

### 3.2. Response Surface Optimization

#### 3.2.1. Identification of Main Variables via Plackett–Burman Design (PBD)

A PBD experiment was conducted to determine key variables influencing the mycelial growth of strain S10. The PBD matrix comprised 12 experimental runs, with the dry mycelial weight as the response variable (Table 1). Based on a preliminary single-factor test, corn flour (X_1_), yeast power (X_2_), NaNO_3_ (X_3_), CaCO_3_ (X_4_), KCl (X_5_), ZnSO_4_·7H_2_O (X_6_), and MnCl_2_·4H_2_O (X_7_) were selected as significant variables for further investigation. The obtained results indicated a variation in the dry mycelial weight ranging from 3.98 to 7.28 g/L, with the highest mycelial biomass yield observed in run 12 (Table 1).

The model adequacy and the significance of the examined variables for the dry mycelial weight were evaluated using ANOVA. As shown in Table 2, the regression model exhibited a significant *p* value of 0.035 (*p* < 0.05), indicating that the model was significant and sufficient. Variables with a confidence level exceeding 95% (*p* < 0.05) were considered statistically significant. As shown in Table 2, corn flour, yeast extract, and ZnSO_4_·7H_2_O were screened as the crucial influencing parameters regarding mycelial biomass, with *p* values at 0.0387, 0.0283, and 0.0108, respectively. Generally, a prob > *F* value less than 0.050 confirms the significance of the model term, further validating its impact on the mycelial dry weight. In contrast, the remaining variables, with *p* values > 0.05, showed no significant effects and were thus excluded from further consideration (Table 2). Additionally, the PBD results revealed positive estimates for corn flour, yeast extract, and ZnSO_4_·7H_2_O, with values of 0.31, 0.34, and 0.46, respectively, reinforcing their substantial contributions to the biomass yield (Table 2). Consequently, corn flour, yeast extract, and ZnSO_4_·7H_2_O were chosen for subsequent optimization using a BBD experiment.

#### 3.2.2. Determination of Central Point Levels by Steepest Climbing Test

To facilitate subsequent response surface analysis, a steepest climbing test was conducted to advance toward the maximum response region of each key parameter. In the steepest climbing assay, the concentration ranges of corn flour, yeast extract, and ZnSO_4_·7H_2_O were chosen as 1.5 g/L, 1.5 g/L, and 0.015 g/L, respectively (Table 3). All other ingredients were maintained at the combinations yielding the maximum mycelial dry weight in the PDB (run 12). The steepest climbing assay design and the corresponding response data are shown in Table 3. The findings indicated that the mycelial dry weight displayed an initial increase followed by a decrease, demonstrating the reliability of the experimental design. Specifically, the mycelial dry weight reached the maximum value of 13.2 g/L in the fourth experiment. Consequently, the proportional composition of the fermentation medium used in treatment 4 was selected as the basal concentration for the subsequent BBD optimization.

#### 3.2.3. Box–Behnken Design (BBD)

Following the steepest climbing assay, a BBD assay was carried out with treatment 4 as the central point. Corn flour (A), yeast extract (B), and ZnSO_4_·7H_2_O (C) were chosen as independent factors in a three-factor, three-level BBD experiment. Table 4 presents the BBD matrix, with each parameter tested at three levels (−1, 0, and 1). Seventeen experimental trials were designed in total, and each trial was carried out in triplicate. The mycelial biomass response (Y) was used as the experimental index. The following quadratic polynomial equation was obtained by performing quadratic multiple regression analysis on the data:Y = +3.19 + 1.35 × A + 1.23 × B + 1.53 × C − 0.77 × AB + 0.35 × AC − 0.25 × BC + 0.99 × A^2^ + 1.32 × B^2^ + 0.61 × C^2^,
which was applied to estimate the predicted production based on the levels of corn flour (A), yeast extract (B), and ZnSO_4_·7H_2_O (C).

The experimental results obtained from the BBD were statistically analyzed through AVOVA to validate the significance of the regression model. Corn flour, yeast extract, and ZnSO_4_·7H_2_O were found to be significant variables, with *p* values less than 0.0001 (Table 5). Statistical evaluation (*p* < 0.01) suggested that the model was reliable. The model exhibited a good fit to the actual data, evidenced by a correlation coefficient (R^2^) of 0.9815. Consequently, the test results can be interpreted through this equation. From Table 5, it is obvious that the model demonstrates high statistical significance (*p* < 0.001), while the lack-of-fit term is not significant (*p* > 0.05), implying that the experimental data are in strong agreement with the predicted responses. The model’s R^2^ is 0.9815, and the adjusted R^2^ is 0.9576, suggesting that the model fits well. The model can explain 95.76% of the variability in the response values. The model terms A, B, C, AB, A^2^, B^2^, and C^2^ were identified as significant terms (Table 5). Moreover, model precision was demonstrated by a low coefficient of variation (C.V. = 9.42%) and an adequate precision ratio of 18.274, substantially exceeding the recommended threshold of 4.0 for satisfactory precision (Table 5).

The three statistically significant variables were subjected to further analysis using the RSM to ascertain the formulation of the optimal fermentation medium via building a mathematical model. The RSM findings suggested that corn flour, yeast extract, and ZnSO_4_·7H_2_O were sufficient to model the response. The constructed 3D response surface plots and 2D contour plots further demonstrated that mycelial biomass yields are predominantly determined by various combinations of two factors, i.e., between corn flour and yeast extract. The interaction between corn flour and ZnSO_4_·7H_2_O, as well as yeast extract (Figure 3A,D), corn flour, and ZnSO_4_·7H_2_O (Figure 3B,E) and yeast extract and ZnSO_4_·7H_2_O (Figure 3C,F), was assessed. The three parameters as optimized by the RSM were as follows: 28.03 g/L for corn flour, 19.99 g/L for yeast extract, and 0.030 g/L for ZnSO_4_·7H_2_O.

### 3.3. Assessment of the Model

Based on the BBD and RSM results, the model predicted a maximum mycelial dry weight of 7.98 g/L, with the following calculated values for the parameters: 28.03 g/L of corn flour, 19.99 g/L of yeast extract, and 0.030 g/L of ZnSO_4_·7H_2_O. To validate the model, strain S10 was fermented in the optimized medium and the mycelial dry weight was quantified. The mycelial dry weight of *S. pratensis* S10 was 8.12 g/L under the optimized conditions, showing strong concordance with the RSM model predictions (Figure 4A). Consequently, these findings confirm that the quadratic polynomial response model provides an effective representation of the expected optimization. Moreover, under optimal conditions, the methanol extracts of the mycelium exhibited higher inhibitory activity against *F. graminearum*, reaching 82.50% (Figure 4B).

### 3.4. Optimization of S10 Enhanced Biocontrol Efficacy Against Fusarium Head Blight

In a growth chamber assay, a control efficacy experiment was carried out using the optimized fermentation broth of S. pratensis S10 against Fusarium head blight. The results demonstrated that the spore suspension of S10 fermented in the optimized medium effectively reduced the diseased spike rate and disease index by 79.75% and 74.70%, respectively (Table 6), compared to the control (CK). Furthermore, when compared to the initial medium, the optimized medium achieved reductions in the diseased spike rate and disease index of 39.24% and 31.17%, respectively (Table 6).

### 3.5. Inactivation of tetR Enhanced Mycelial Growth and Antifungal Activity of S10

We enhanced the antifungal activity of the wild-type strain S10 through genetic modification, constructing a negative regulator gene *tetR* via double-crossover homologous recombination. In this process, the *tetR* coding sequence was replaced with a gentamicin resistance gene (Figure 5A). The mutant Δ*tetR* was successfully constructed, as displayed in Appendix A. As shown in Figure 5B, Δ*tetR* grew faster than the wild-type strain. In liquid medium, Δ*tetR* still showed the fastest growth. The mycelial biomass of Δ*tetR* was greater than that of the wild-type strain throughout the 6–120 h period (Figure 5C). Importantly, Δ*tetR* exhibited higher antifungal activity against *F. graminearum* than the wild-type strain S10 (Figure 5D). To complement *tetR* deletion, a DNA fragment containing *tetR* was cloned into pSETC to generate pSETC-*tetR*, which was subsequently introduced into Δ*tetR* using *E. coli* ET12567. PCR amplification of the apramycin resistance gene was performed to select the complementation strain com-*tetR*. There was no significant difference in mycelial growth or antifungal activity between the wild-type strain S10 and the complemented com-*tetR* strain (Figure 5B–D).

### 3.6. Optimized Medium Increased Antifungal Activity of ΔtetR Against F. graminearum

We evaluated whether the enhanced antifungal production achieved by integrated fermentation optimization and genetic engineering could improve the mycelial biomass and antifungal activity. Figure 6A indicates that the mycelial dry weight of Δ*tetR* was 11.53 g/L under the optimized medium, showing a 4.24-fold difference from that of the wild-type strain S10. The results also showed that the mycelial extract from the Δ*tetR* cultured in the optimized medium demonstrated significantly greater inhibition of *F. graminearum* mycelial growth compared to the wild-type strain S10 grown in the unoptimized GS medium (Figure 6B).

## 4. Discussion

Extensive studies have demonstrated that *Streptomyces* spp. are effective BCAs that exhibit excellent suppressive activity against multiple phytopathogens [30,31,32,33]. The production of secondary metabolites in *Streptomyces* spp. is highly dependent on both the strain and the composition of the fermentation medium [34,35,36], leading to considerable variations in metabolite profiles across different culture media. Our previous study confirmed that *S. pratensis* S10 can suppress mycelial growth and DON production in *F. graminearum* through the secretion of antifungal secondary metabolites [11]. However, the practical application of *S. pratensis* S10 has been impeded due to its low antifungal substance production. This poses a major challenge for large-scale industrial production. To overcome this limitation, it is necessary to optimize the fermentation broth of *S. pratensis* S10. Therefore, in this study, we optimized the fermentation medium for strain S10 using a systematic approach combining single-factor experiments and the PBD, BBD, and RSM. This strategy aimed to enhance the biomass yield and maximize antifungal metabolite production, thereby facilitating the further development and application of *S. pratensis* S10 as a sustainable biocontrol agent.

The effective selection of carbon and nitrogen sources, essential in optimizing microbial growth and reproduction, is a key part of culture medium optimization in industrial production due to strain-specific nutritional requirements [37,38,39], thereby boosting the yields and quality of bioactive compounds. According to previous studies, glucose, maltose, sucrose, corn flour, fructose, galactose, oats, and soluble starch were selected as potential carbon sources. The results from single-factor assays indicated that corn flour was identified as the most suitable carbon source for the growth and antifungal activity of strain S10. Yang et al. (2025) reported that millet serves as the best carbon source for the growth and reproduction of *Streptomyces* sp. KN37 [25]. These results suggest that the optimal carbon source for different *Streptomyces* species to produce secondary metabolites with bioactive activity often varies from strain to strain. Moreover, our study showed that fructose had a negative effect on antifungal activity. A possible explanation for this phenomenon is that fructose may cause catabolite repression, in which the production of enzymes for secondary metabolite biosynthesis might be inhibited [40]. In the nitrogen source screening, yeast extract and NaNO_3_ were found to promote high antifungal activity and mycelial dry weights; these findings were consistent with the results reported by Liu et al. [26]. Minerals such as Na^+^, K^+^, Ca^2+^, and Mg^2+^ are essential for microbial metabolism and growth, serving as enzyme activators [26]. Inorganic salts supply critical mineral elements that support microbial development and have demonstrated efficacy in suppressing various diseases [26]. For instance, Wang et al. (2023) reported that the supplementation of CaCl_2_ to the culture medium significantly enhanced the production of lipopeptide herbicolin in *Pantoea agglomerans* ZJU23 [8]. Trace elements also play a crucial role in secondary metabolism. In this study, the addition of CaCO_3_, MnCl_2_·4H_2_O, and ZnSO_4_·7H_2_O to the fermentation medium markedly improved the antifungal activity, underscoring the significance of these elements in optimizing microbial secondary metabolism and bioactive compound synthesis.

Secondary metabolite biosynthesis by microorganisms demonstrates a strong dependence on both strain-specific characteristics and the culture medium composition. Microbial fermentation processes are inherently complex, dynamic, and challenging to control, with even minor modifications to medium formulations potentially exerting significant effects on the production of antifungal substances and altering key metabolic pathways. The RSM provides a robust framework for fermentation medium optimization [25], thereby increasing secondary metabolite synthesis and enhancing bioactive compound yields. The PBD experiment was initially employed to identify major determinants influencing metabolite synthesis [8]. In this study, corn flour, yeast extract, and ZnSO_4_·7H_2_O emerged as significant factors, enhancing both biomass accumulation and antifungal activity, warranting their selection for subsequent optimization. These findings corroborate previous work by Shakeel et al. (2016), confirming that the RSM is an effective approach in fermentation parameter optimization due to its rapid testing cycle, high precision, and reduced experimental requirements [39]. The RSM-derived optimization model successfully identified the fermentation medium composition yielding maximal activity. Validation experiments demonstrated excellent agreement between the predicted value of 7.98 g/L and the observed mycelial dry weight of 8.12 g/L. Furthermore, a significant improvement in the antifungal activity of strain S10 was achieved through optimization.

The majority of metabolic engineering strategies aimed at enhancing the antifungal activity of BCAs focus on improving the production of bioactive compounds [41]. One such strategy involves deleting negative regulator genes. For instance, the deletion of genes involved in avermectin biosynthesis obviously enhanced oligomycin production in *S. avermitilis* [42]. The genome of *S. pratensis* S10 harbors 30 secondary metabolic biosynthesis gene clusters, including those encoding PKSs, nucleosides, non-ribosomal peptides, siderophores, siderophores, and other compounds [11]. To develop a strain that produces a high yield of bioactive substances, here, we selected a negative transcript repressor, *tetR*, belonging to the TetR family of transcriptional regulators, in the predicted bioactive substance biosynthesis gene cluster. TetR-family regulators are generally identified as negative regulators of antibiotic production in *Streptomyces* spp. [43]. Consistently, our results showed that *tetR* negatively regulated the mycelial growth and antifungal activity of strain S10. Disruption of the *tetR* gene in strain S10 markedly increased the antifungal activity. Furthermore, the mutant exhibited faster growth and sporulation, along with increased fermentation.

## 5. Conclusions

In summary, enhancements in the mycelial growth and antifungal activity of strain S10 were achieved by integrating fermentation optimization and strain engineering. We successfully optimized the composition of the fermentation medium to enhance both the biomass yield and the production of antifungal metabolites in *S. pratensis* S10. The optimized fermentation medium led to a significant increase in the biocontrol efficacy of *S. pratensis* S10 against FHB. The Δ*tetR* mutant under the optimized fermentation medium evidently increased the antifungal activity against *F. graminearum*. These findings provide a solid foundation for the future development of bioactive compounds derived from S10 and offer new insights and potential applications for *S. pratensis* S10 in sustainable agriculture.

## Figures and Tables

**Figure 1 microorganisms-13-01943-f001:**
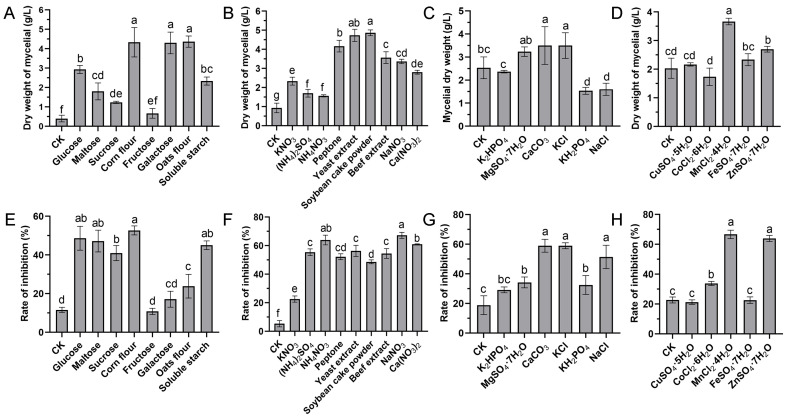
Effects of diverse medium compositions on the mycelial biomass and antifungal activity of *S. pratensis* S10. (**A**,**E**) Carbon sources. (**B**,**F**) Nitrogen sources. (**C**,**G**) Mineral salts. (**D**,**H**) Trace elements. The mycelial dry weight represented biomass, while the rate of inhibition indicated antifungal compound generation. Data are expressed as mean ± s.d. of three replicates. Significant differences (*p* < 0.05) among treatments are denoted by different lowercase letters according to Duncan’s test.

**Figure 2 microorganisms-13-01943-f002:**
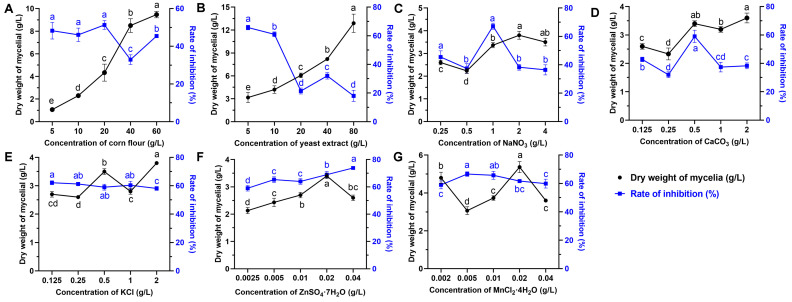
Effects of nutrient sources at different concentrations on the biomass yield and antifungal metabolite production of *S. pratensis* S10. (**A**) Corn flour. (**B**) Yeast powder. (**C**) NaNO_3_. (**D**) CaCO_3_. (**E**) KCl. (**F**) ZnSO_4_·7H_2_O. (**G**) MnCl_2_·4H_2_O. The dry weight of mycelium was used as an indication of biomass (corresponds to the left X axis), and the rate of inhibition was used as an indication of antifungal metabolite production (corresponds to the right Y axis). Data are expressed as mean ± s.d. of three replicates. Different lowercase letters indicate a significant difference at the *p* < 0.05 level by Duncan’s new multiple-range test.

**Figure 3 microorganisms-13-01943-f003:**
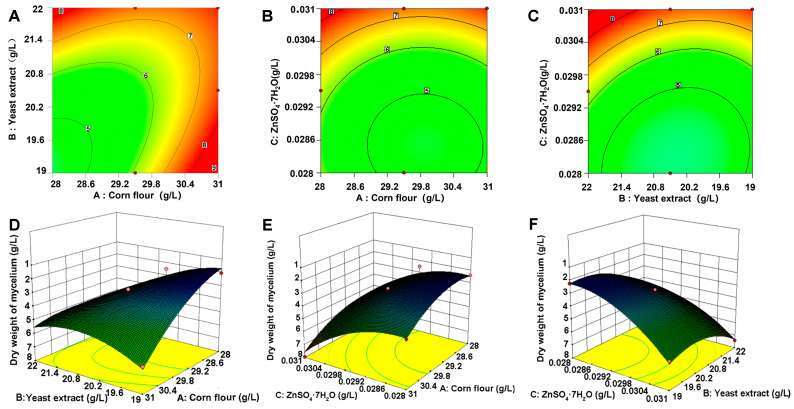
Two-dimensional contour and 3D response surfaces for assessment of effects. (**A**) Interaction of corn flour and yeast extract. (**B**) Interaction of corn flour and ZnSO_4_·7H_2_O. (**C**) Interaction of yeast extract and ZnSO_4_·7H_2_O. Response surface plot illustrating interactive effects on mycelial dry weight. (**D**) Corn flour and yeast extract. (**E**) Corn flour and ZnSO_4_·7H_2_O. (**F**) Yeast extract and ZnSO_4_·7H_2_O. (**A**–**C**) represent the two-dimensional contour plots and (**D**–**F**) represent the three-dimensional response surfaces. The numbers 5–9 in the figures indicate the mycelium dry weight at this location. Different colors represent different values of the response variable.

**Figure 4 microorganisms-13-01943-f004:**
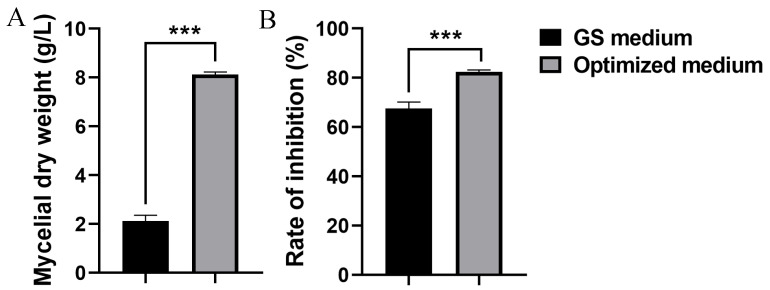
Evaluation of the statistically optimized medium versus the initial medium (GS medium). (**A**) Mycelial dry weight. (**B**) Inhibition rate. Data are expressed as mean ± s.d. of three replicates. *** indicates statistically significant value (*p* < 0.001) according to Student’s *t* test.

**Figure 5 microorganisms-13-01943-f005:**
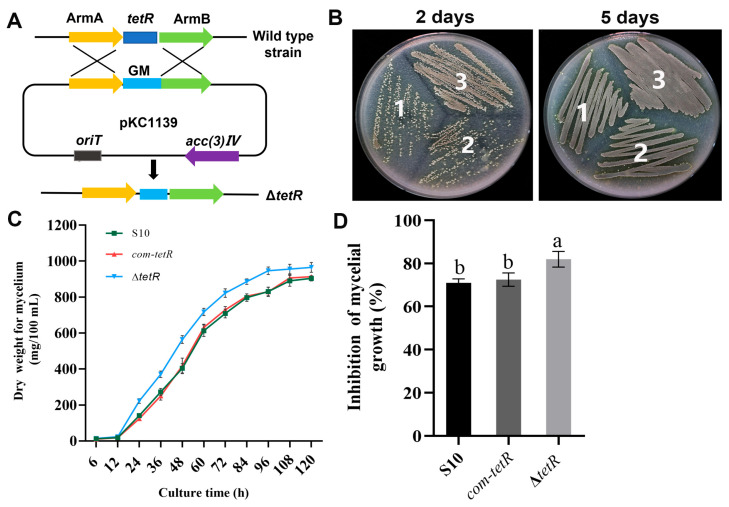
The effects of *tetR* on the mycelial growth and antifungal activity of *S. pratensis* S10. (**A**) Replacement of *tetR* in *S. pratensis* S10. (**B**) Morphological development of each strain on mannitol soya flour (MS) medium containing 20 g soya flour, 20 g mannitol, and 15 g agar per liter. 1, wild-type *S. pratensis* S10; 2, Δ*tetR*; 3, com-*tetR*. (**C**) Mycelial biomass of each strain during culture for 120 h. (**D**) Rate of inhibition. Values are shown as mean ± s.d. of three independent biological replicates. Significant differences in distinct groups are indicated by distinct letters (*p* < 0.05, ANOVA, Duncan’s test).

**Figure 6 microorganisms-13-01943-f006:**
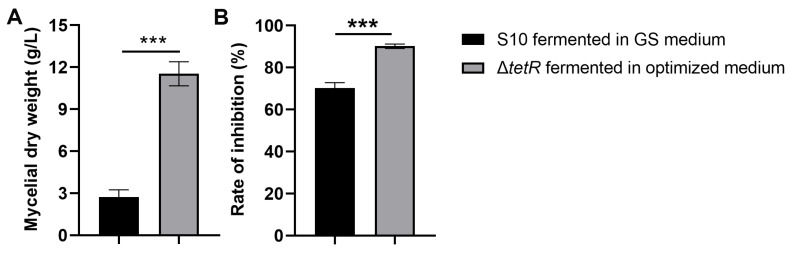
Mycelial dry weight (**A**) and antifungal activity (**B**) of strain S10 was enhanced after fermentation optimization and strain engineering. Data are expressed as mean ± s.d. of three replicates. *** indicates statistical significance based on Student’s *t* test (*p* < 0.001).

**Table 1 microorganisms-13-01943-t001:** Design and corresponding results of PBD.

Run	Code Number							Dry Mycelial Weight (g/L)
	X_1_	X_2_	X_3_	X_4_	X_5_	X_6_	X_7_	
1	−1	−1	−1	1	−1	1	1	4.42
2	1	−1	−1	−1	1	−1	1	4.91
3	1	−1	1	1	1	−1	−1	4.25
4	1	−1	1	1	−1	1	1	4.54
5	−1	1	−1	1	1	−1	1	4.31
6	−1	1	1	−1	1	1	1	4.90
7	−1	1	1	1	−1	−1	−1	4.67
8	−1	−1	−1	−1	−1	−1	−1	3.98
9	1	1	−1	−1	−1	1	−1	6.00
10	−1	−1	1	−1	1	1	−1	4.00
11	1	1	1	−1	−1	−1	1	4.27
12	1	1	−1	1	1	1	−1	7.28

**Table 2 microorganisms-13-01943-t002:** Main results of PBD experiments.

Source	Sum of Squares	df	Coefficient Estimate	Mean Square	*F* Value	Prob > *F*
Model	7.05	7	4.81	1.01	8.11	0.0305 *
Corn flour	1.14	1	0.31	1.14	9.19	0.0387 *
Yeast extract	1.40	1	0.34	1.40	11.28	0.0283 *
NaNO_3_	0.80	1	−0.26	0.80	6.45	0.0640
CaCO_3_	0.041	1	−0.058	0.041	0.33	0.5970
KCl	0.24	1	0.14	0.24	1.94	0.2361
ZnSO_4_·7H_2_O	2.52	1	0.46	2.52	20.30	0.0108 *
MnCl_2_·4H_2_O	0.91	1	−0.27	0.91	7.31	0.0539
R-Squared	0.50	4				

* indicates a significant difference at the *p* < 0.05 level.

**Table 3 microorganisms-13-01943-t003:** Design and results of the steepest climb experiment.

Run	Corn Flour (g/L)	Yeast Extract (g/L)	ZnSO_4_·7H_2_O (g/L)	Dry Weight of Mycelium (g/L)
1	25	6.25	0.025	6.42
2	26.5	7.5	0.0265	7.52
3	28	8.75	0.028	7.18
4	29.5	10.0	0.0295	13.2
5	31	11.25	0.031	8.25
6	32.5	12.5	0.0325	8.62
7	34	13.75	0.034	10.1

**Table 4 microorganisms-13-01943-t004:** The experimental framework and findings of the BBD.

Run	Code Number			Actual Value (g/L)	Predicted Value (g/L)
	A	B	C		
1	0	0	0	3.03	3.19
2	1	−1	0	6.13	6.38
3	1	0	−1	4.34	4.25
4	−1	0	0	2.48	2.84
5	0	0	0	3.13	3.19
6	0	1	−1	4.75	6.08
7	1	0	1	7.98	8.01
8	0	0	0	3.93	3.19
9	0	1	1	7.42	9.63
10	0	−1	1	5.91	5.67
11	−1	−1	0	2.05	2.15
12	−1	0	−1	2.39	2.27
13	0	0	0	3.00	3.19
14	−1	1	0	6.5	6.16
15	0	−1	−1	2.24	2.11
16	0	0	0	3.06	3.19
17	1	1	0	7.51	7.31

**Table 5 microorganisms-13-01943-t005:** Design and results of BBD.

Source	Sum of Squares	df	Mean Square	*F* Value	Prob > *F*
Model	65.50	9	7.28	41.16	<0.0001 ***
A—Corn starch	12.17	1	12.17	68.84	<0.0001 ***
B—Yeast extract	12.13	1	12.13	68.60	<0.0001 ***
C—ZnSO_4_·7H_2_O	13.08	1	13.08	73.99	<0.0001 ***
AB	2.36	1	2.36	13.33	0.0082 **
AC	0.27	1	0.27	1.54	0.2541
BC	0.25	1	0.25	1.41	0.2732
A^2^	3.77	1	3.77	21.34	0.0024 **
B^2^	6.61	1	6.61	37.39	0.0005 ***
C^2^	1.23	1	1.23	6.93	0.0338 *
Residual	1.24	7	0.18	–	–
Lack of Fit	0.62	3	0.21	1.32	0.3845, not significant
Pure Error	0.62	4	0.16	–	–
Cor. Total	66.73	16	–	–	–
Std. Dev.	0.45	–	R-Squared	–	0.9815
Mean	4.46	–	Adj. R-Squared	–	0.9576
C.V.%	9.42	–	Pred. R-Squared	–	0.7535
PRESS	16.45	–	Adeq. Precision	–	18.274

Statistical significance is denoted by * (*p* < 0.05), ** (*p* < 0.01), and *** (*p* < 0.001).

**Table 6 microorganisms-13-01943-t006:** The control efficacy of the optimized fermentation medium on Fusarium head blight.

Treatment	Disease Spike Rate (%)	Disease Index	Control Efficacy (%)
CK	100.00 ± 00 a	90.95 ± 6.40 a	—
Initial medium	33.33 ± 4.26 b	33.43 ± 2.81 b	63.25 ± 3.09 a
Optimized medium	20.25 ± 0.75 c	23.01 ± 2.82 c	74.70 ± 3.10 b

Values are expressed as mean ± s.d. from three independent biological replicates. Significant differences (*p* < 0.05, ANOVA, Duncan’s test) are indicated by distinct letters.

## Data Availability

The original contributions presented in this study are included in the article/Appendix A. Further inquiries can be directed to the corresponding authors.

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
