# Peer review of "Enhancement of Mycelial Growth and Antifungal Activity by Combining Fermentation Optimization and Genetic Engineering in Streptomyces pratensis S10"

_microorganisms, 2025, doi:10.3390/microorganisms13081943_

Round 1

Reviewer 1 Report

Comments and Suggestions for Authors

1. In the introduction, it would be useful to mention exactly why you are measuring mycelial growth throughout the experiments. My impression is that mycelial growth in this Streptomyces strain correlates with cell differentiation and activation of secondary metabolism, which produces the biocontrol molecules that are interesting. I believe this point could be made more clearly in order to help readers understand the purpose of some of the experiments described in the paper. 

2. Regarding the tetR gene described in the paper, please provide the appropriate gene ID/number. This is a publicly available genome (NCBI assembly GCA_016804005.1) and the gene identifier should be present. Is it actually a tetracycline resistance gene? It would also be helpful to add a few lines to the manuscript as to why you decided to choose to target this gene product, as the rationale is hidden in the discussion.

3. In addition, a methanol extract was made from tetR-KO cells for testing of inhibition, but was there any HPLC or LC-MS analysis of the fraction carried out? I am asking because if you really did KO a global transcriptional regulator, your data suggests activation of new BGCs that may synthesize products that contribute to the results reported in Figures 4 and 5. The increase in mycelial biomass with the gene KO also suggests that engineered cells may differentiate more efficiently than WT.

4. In section 2.6, what polymerase was used to amplify the DNA parts from genomic DNA? I am asking because Strep is a high-GC organism that is difficult to amplify with standard polymerases and this could be helpful information to report.

5. I would consider moving Supplemental Figure 1 into the main text, as the changes in fungal inhibition with increasing amounts of nutrient sources is interesting. If not, perhaps a small section in the discussion regarding the inverse relationship of nutrient vs antifungal activity is warranted. Why do you think that this trend was observed? Is it just a matter of too much of a nutrient source disrupting the fermentation process, or could this be a physiological effect in the bacteria, like catabolic repression?

6. In Supplemental Figure 2, if possible, please replace the gel images with ones that do not show the saturation of DNA bands (the red lines obscuring some of the image). This is usually easily turned off in the gel imager software settings. Also, in Figure S2E, the overloading of the 3 GM lanes makes it impossible to see the DNA ladder and if there are any bands in the Am lanes (image is highly saturated). If possible, replace this image with one with reduced DNA in the GM lanes in order to allow for better exposure of the entire gel.

Author Response

Comment1: In the introduction, it would be useful to mention exactly why you are measuring mycelial growth throughout the experiments. My impression is that mycelial growth in this Streptomyces strain correlates with cell differentiation and activation of secondary metabolism, which produces the biocontrol molecules that are interesting. I believe this point could be made more clearly in order to help readers understand the purpose of some of the experiments described in the paper. 

Response1: Thanks for your suggestion. We mentioned why we measuring mycelial growth in the Materials and Methods section 2.3 (lines 143-149). Generally, in non-growth associated fermentation, secondary metabolites production was substantially proportional to the quantity of biomass. Therefore, it is essential to increase the total biomass of microorganisms, especially S. pratensis S10, since its antifungal metabolites were produced inside the cells. Taking in account the above theory, we focused on optimization of medium components, seeking a substantial increase in biomass without reducing the production of secondary metabolites. Therefore, mycelial dry weight was utilized as the response in subsequent experiment designs.

Comment2: Regarding the tetR gene described in the paper, please provide the appropriate gene ID/number. This is a publicly available genome (NCBI assembly GCA_016804005.1) and the gene identifier should be present. Is it actually a tetracycline resistance gene? It would also be helpful to add a few lines to the manuscript as to why you decided to choose to target this gene product, as the rationale is hidden in the discussion.

Response2: Thanks for your suggestion very much. We provided the tetR gene ID (Line 219). It is not a tetracycline resistance gene. And, we also added a few lines to the discussion as why we decided to select the gene (lines 526-529).

Comment3: In addition, a methanol extract was made from tetR-KO cells for testing of inhibition, but was there any HPLC or LC-MS analysis of the fraction carried out? I am asking because if you really did KO a global transcriptional regulator, your data suggests activation of new BGCs that may synthesize products that contribute to the results reported in Figures 4 and 5. The increase in mycelial biomass with the gene KO also suggests that engineered cells may differentiate more efficiently than WT.

Response3: Thanks for your question very much. Currently, we did not carry out the HPLC or LC-MS analysis. We just compared the antifungal activity between the WT and the mutant. We thank the constructive comments you provided will be of great help to our future research. Our currently work focused on analyzing how this gene regulates the gene cluster, and we will conduct in-depth studies on this aspect of the work. Thank you again.

Comment4: In section 2.6, what polymerase was used to amplify the DNA parts from genomic DNA? I am asking because Strep is a high-GC organism that is difficult to amplify with standard polymerases and this could be helpful information to report.

Response4: Thanks for asking and the suggestion help us improve the quality of manuscript. Phusion DNA polymerase (New England Biolabs, Inc., United States). We have provided the information in section 2.6 (line 229).

Comment5: I would consider moving Supplemental Figure 1 into the main text, as the changes in fungal inhibition with increasing amounts of nutrient sources is interesting. If not, perhaps a small section in the discussion regarding the inverse relationship of nutrient vs antifungal activity is warranted. Why do you think that this trend was observed? Is it just a matter of too much of a nutrient source disrupting the fermentation process, or could this be a physiological effect in the bacteria, like catabolic repression?

Response5: Thanks for you suggestion. We have moved the Supplementary Figure 1 into the main text (Figure 2 in revised manuscript). We also discuss the inverse relationship of nutrient vs antifungal activity is warranted. We think this phenomenon is might be a physiological effect in bacteria, catabolites repression, in which the production of enzymes for secondary metabolites biosynthesis might be inhibited (lines 485-488).

Comment6: In Supplemental Figure 2, if possible, please replace the gel images with ones that do not show the saturation of DNA bands (the red lines obscuring some of the image). This is usually easily turned off in the gel imager software settings. Also, in Figure S2E, the overloading of the 3 GM lanes makes it impossible to see the DNA ladder and if there are any bands in the Am lanes (image is highly saturated). If possible, replace this image with one with reduced DNA in the GM lanes in order to allow for better exposure of the entire gel.

Response6:  Thanks for your suggestion. We have replaced the gel images. Thank you help us improve the quality of images again.

Reviewer 2 Report

Comments and Suggestions for Authors

In this manuscript, the authors evaluated the antifungal activity of Streptomyces pratensis S10 by integrating fermentation optimization using Plackett-Burman Design (PDB) and Box-Behnken design (BBD) to obtain an optimized medium for inhibitory activity and mycelial biomass for its commercialization, in addition mutant strains were obtained with increased results.  In this sense, the results show important findings using some microbiological and statistical methods analysis to determine optimal conditions. The research work is very interesting; all results are well-performed.

Comments:

1.- Line 70, edit Streptomyce spp.

2.- Line 129, please describe the component of GS medium to understand the concentration of these components.

3.-Line 197, please describe the conditions of growth for the wheat plants at 10% anthesis stage used.

4.-Line 201, please describe the fungal strain used in this experiment.

5.-Line 208, Please describe in detail the conditions for PCR, conjugation and culture medium used for mutant selection. Also describe in detail the tetR gene to understand why this gene was selected.

6.-In discussion, a detail function of tetR gene is missing to understand these results for the interpretation. Why the tetR gene was selected?

7.-An discussion of repression catabolic for carbon sources is necessary to understand the interpretation of these results and other components of GS medium.  

Author Response

Comment 1: Line 70, edit Streptomyce spp.

Response 1: Thanks for your pointing out. We have changed the “Streptomyce” to “Streptomyces” in line 70.

Comment 2: Line 129, please describe the component of GS medium to understand the concentration of these components.

Response 2: Thank for your pointing out. We have described the component of GS medium in lines 129-130.

Comment 3: Line 197, please describe the conditions of growth for the wheat plants at 10% anthesis stage used.

Response 3: Thanks for pointing out. We have described the conditions of growth for the wheat plants at 10% anthesis stage used (lines 205-207).

Comment 4: Line 201, please describe the fungal strain used in this experiment.

Response 4: Thanks for pointing out. We added the fungal strain (line 211: F. graminearum) used in this experiment.

Comment 5: Line 208, Please describe in detail the conditions for PCR, conjugation and culture medium used for mutant selection. Also describe in detail the tetR gene to understand why this gene was selected.

Response 5: Thanks for your suggestion. We described in detail the conditions for PCR (lines 228-232); conjugation and culture medium for mutant selection are GS and MS medium, respectively (line 226 and line 227). We added the detail to describe why this gene was selected (lines 219-220). And, we also added a few lines to the discussion as why we decided to select the gene (lines 526-529).

Comment 6: In discussion, a detail function of tetR gene is missing to understand these results for the interpretation. Why the tetR gene was selected?

Response 6: Thanks for your suggestion. We added the detail function of tetR gene in the discussion (lines 526-529). tetR gene is a negative transcript repressor in the predicted bioactive substance biosynthesis gene cluster. Thus, we selected the tetR gene.

Comment 7: An discussion of repression catabolic for carbon sources is necessary to understand the interpretation of these results and other components of GS medium. 

Response 7:  Thanks for your suggestion. We have added the repression catabolic for carbon sources in the Discussion section (lines 485-488). 

Reviewer 3 Report

Comments and Suggestions for Authors

Enhancement of Mycelial Growth and Antifungal Activity by Combining Fermentation Optimization and Genetic Engineering in Streptomyces pratensis S10

Lifang Hu, Yan Sun, Ruimin Jia, Xiaomin Dong, Xihui Shen, and Yang Wang

     This manuscript covers the following topics: microbiology, molecular biology, fermentation process optimization, and gene engineering. The article has a clear structure, and its objective is well reflected in the content. The research focuses on enhancing the antifungal activity of Streptomyces pratensis S10 through fermentation optimization and genetic engineering. Results from single-factor experiments revealed seven significant composition parameters: corn flour, yeast extract, NaNO₃, CaCO₃, Kâ‚‚HPOâ‚„, KCl, ZnSOâ‚„·7Hâ‚‚O, and MnClâ‚‚·4Hâ‚‚O. The study showed that the tetR gene negatively impacted the production of bioactive compounds and that the improved medium promoted favorable growth conditions for S. pratensis S10. These findings lay the groundwork for the large-scale production of bioactive metabolites from S. pratensis S10 and offer a promising, sustainable approach to managing Fusarium head blight in agriculture.

Comments for the authors:

  • Correct the writing of the chemical formula: (NH4)2SO4 in Figure 1 (B, F).
  • Correct the writing of the chemical formula: MgSO47H2O in Figure 1 (C, G).
  • Correct the writing of the chemical formula: CoCl26H2O in Figure 1 (D, H).
  • Correct the writing of the chemical formula: ZnSO47H2O in Supplementary Figure 1 (F).
  • Correct the spelling of the title of the Table 2, „Main effect of analysis of PBD experiments“.
  • Could you please provide the composition of the MS medium used in the Figure 4B?
  • Correct the spelling in the sentence, „showing 4.24-fold than that of wild type strain S10“ in the paragraph 424.
  • Correct the spelling of the sentence, „The results from single-factor assays indicated that corn flour was identified as the most suitable carbon sources for the growth and antifungal activity of strain S10“.
  • Correct the spelling of the word „engineering“ (the „e“ should not be bold) in the title of the Figure 5 in paragraph 431.
  • I could not find citation for Yang et al. in the list of references, „Yang et al. (2025) reported that millet serves as the best carbon source for the growth and reproduction of Streptomyces KN37“ in the paragraph 456.
  • I did not find the Wang et al. citation in the list of references, „For instance, Wang et al. (2023) reported that the supplementation of CaCl2 to the culture“ in the paragraph 465.
  • I did not find the Shakeel et al. citation in the list of references, „These findings corroborate previous work by Shakeel et al. (2016)“ in the paragraph 482.
  • I did not find the mandatory „Conclusions“ section according to the requirements of the journal „Microorganisms“.
  • Check the proofreading of the titles of Supplementary Tables 1,2, and 3.
  • Correct the proofreading of one reference in the Supplementary material.
  • Can Streptomyces pratensis S10 be used in industrial biotechnology applications, and, if so, how?

Author Response

Comment 1: Correct the writing of the chemical formula: (NH4)2SOin Figure 1 (B, F).

Response 1: Thanks for your pointing out. We have changed the writing of the chemical formula: (NH4)2SOin Figure 1 (B, F).

Comment 2: Correct the writing of the chemical formula: MgSO47H2O in Figure 1 (C, G).

Response 2: Thanks for your pointing out. We have changed the writing of the chemical formula: MgSO47H2O in Figure 1 (C, G).

Comment 3:Correct the writing of the chemical formula: CoCl26H2O in Figure 1 (D, H).

Response 3: Thanks for your pointing out. We have changed the writing of the chemical formula: CoCl26H2O in Figure 1 (D, H).

Comment 4: Correct the writing of the chemical formula: ZnSO47H2O in Supplementary Figure 1 (F).

Response 4: Thanks for your pointing out. We have changed the writing of the chemical formula: ZnSO47H2O in Supplementary Figure 1 (F) (Figure 2 in revised manuscript).

Comment 5: Correct the spelling of the title of the Table 2, „Main effect of analysis of PBD experiments”.

Response 5: Thanks for your pointing out. We have corrected the spelling of the title of the Table 2. Thank you again.

Comment 6: Could you please provide the composition of the MS medium used in the Figure 4B?

Response 6: Thank for your suggestion. We have provided the composition of the MS medium, which contains 20 g soya flour, 20 g mannitol, and 15 g agar per liter (line 440-441).

Comment 7: Correct the spelling in the sentence, „showing 4.24-fold than that of wild type strain S10“ in the paragraph 424.

Response 7: Thanks for your pointing out. We have corrected the spelling of the strain (line 450). Thank you again.

Comment 8: Correct the spelling of the sentence, „The results from single-factor assays indicated that corn flour was identified as the most suitable carbon sources for the growth and antifungal activity of strain S10“.

Response 8: Thanks for your pointing out. We have corrected the spelling of strain in line 481.

Comment 9: Correct the spelling of the word „engineering“ (the „e“ should not be bold) in the title of the Figure 5 in paragraph 431.

Response 9: Thanks for pointing out. We have corrected the spelling of the word “engineering” in line 457.

Comment 10: I could not find citation for Yang et al. in the list of references, „Yang et al. (2025) reported that millet serves as the best carbon source for the growth and reproduction of Streptomyces KN37“ in the paragraph 456.

Response 10: Thanks for asking. The citation for Yang er al. in the list of reference of [25].

Comment 11: I did not find the Wang et al. citation in the list of references, „For instance, Wang et al. (2023) reported that the supplementation of CaCl2 to the culture“ in the paragraph 465.

Response 11: Thanks for asking. The Wang et al. citation in the list of reference of [8].

Comment 12: I did not find the Shakeel et al. citation in the list of references, „These findings corroborate previous work by Shakeel et al. (2016)“ in the paragraph 482.

Response 12: Sorry for my mistakes. We have added the Shakeel et al. citation in the list of references of [39].

Comment 13: I did not find the mandatory „Conclusions“ section according to the requirements of the journal „Microorganisms“.

Response 13: Thanks for pointing out. We added the “Conclusion” section in lines 533-542.

Comment 14: Check the proofreading of the titles of Supplementary Tables 1,2, and 3.

Response 14: Thanks for your comments. We have checked the proofreading of the titles of Supplementary Tables 1,2, and 3.

Comment 15: Correct the proofreading of one reference in the Supplementary material.

Response 15: Thanks for your comments. We have corrected the proofreading of one reference in the Supplementary material.

Comment 16: Can Streptomyces pratensis S10 be used in industrial biotechnology applications, and, if so, how?

Response 16: Thanks for your constructive suggestion. S. pratensis S10 is known to have a high control efficiency towards Fusarium graminearum. However, lower production rates and high costs are the main limiting factors for its used in industrial biotechnology applications. Therefore, simple process will likely require further optimization to facilitate its application in the industrial biotechnology in the future.